# Subcellular and supracellular mechanical stress prescribes cytoskeleton behavior in *Arabidopsis* cotyledon pavement cells

Arun Sampathkumar[1,2†], Pawel Krupinski[3†], Raymond Wightman[2], Pascale Milani[4], Alexandre Berquand[5], Arezki Boudaoud[4], Olivier Hamant[4]*, Henrik Jönsson[2,3]*, Elliot M Meyerowitz[2,6]*

[1]Division of Biology and Biological Engineering, California Institute of Technology, Pasadena, United States; [2]Sainsbury Laboratory, University of Cambridge, Cambridge, United Kingdom; [3]Computational Biology and Biological Physics Group, Department of Astronomy and Theoretical Physics, Lund University, Lund, Sweden; [4]Laboratoire de Reproduction et Développement des Plantes, INRA-CNRS-UCBL-ENS Lyon, Lyon, France; [5]Bruker AXS, Bruker Nano GmbH, Mannheim, Germany; [6]Division of Biology and Biological Engineering, Howard Hughes Medical Institute, California Institute of Technology, Pasadena, United States

**Abstract** Although it is a central question in biology, how cell shape controls intracellular dynamics largely remains an open question. Here, we show that the shape of *Arabidopsis* pavement cells creates a stress pattern that controls microtubule orientation, which then guides cell wall reinforcement. Live-imaging, combined with modeling of cell mechanics, shows that microtubules align along the maximal tensile stress direction within the cells, and atomic force microscopy demonstrates that this leads to reinforcement of the cell wall parallel to the microtubules. This feedback loop is regulated: cell-shape derived stresses could be overridden by imposed tissue level stresses, showing how competition between subcellular and supracellular cues control microtubule behavior. Furthermore, at the microtubule level, we identified an amplification mechanism in which mechanical stress promotes the microtubule response to stress by increasing severing activity. These multiscale feedbacks likely contribute to the robustness of microtubule behavior in plant epidermis.

**\*For correspondence:** Olivier. Hamant@ens-lyon.fr (OH); Henrik.Jonsson@slcu.cam.ac.uk (HJ); meyerow@caltech.edu (EMM)

[†]These authors contributed equally to this work

**Competing interests:** The authors declare that no competing interests exist.

**Reviewing editor**: Dominique Bergmann, Stanford University, United States

## Introduction

Epithelia have a crucial role during the development of most multicellular organisms. Consistently, several mechanisms ensure some level of coordination between epithelial cells. In addition to biochemical signals, such as morphogens that diffuse and provide regional coordination across several cell files (*Wolpert, 1969*; *Jonsson et al., 2006*; *Jaeger et al., 2008*; *Wartlick et al., 2009*), mechanical stress also contributes to growth coordination, for instance by synchronizing cell proliferation rate (*Shraiman, 2005*) and orientation (*Thery et al., 2007*), or by prescribing cell polarity (*Asnacios and Hamant, 2012*) and cell fate (*Engler et al., 2006*). In theory, these coordinating mechanisms could lead to relatively homogeneous cell shapes. While this is observed in some classic cases, such as the cells of *Drosophila* ommatidia or *Arabidopsis* petals, most epithelia exhibit variable cell sizes and shapes, demonstrating that each cell retains the ability to regulate its own growth and shape (*Roeder et al., 2010*, *2012*). This heterogeneity has been studied in several systems. In *Drosophila* embryos, stochastic actomyosin-dependent constrictions of cells occur during gastrulation (*Martin et al., 2009*) and dorsal closure (*Solon et al., 2009*), and this stochasticity has been proposed to play a key role in invagination events

**eLife digest** The surfaces of plants are covered in epithelial cells that come in many different shapes, suggesting that individual cells must have some control over their own shape. An unusually shaped epithelial cell is the pavement cell, which looks like a jigsaw puzzle piece and is found in the leaves of many flowering plants. Relatively little was known about the exact contribution of mechanical properties of the wall to this shape. Furthermore, although it was known that parts of pavement cells are rich in microtubules—tubes of protein that act as a scaffold inside the cell— the possibility that shape impacts the behavior of microtubules was not fully addressed.

Now, using a combination of computer modelling and experiments, Sampathkumar et al. reveal that the shape of the pavement cells relies in part on the response of the microtubules to stress. In an individual cell, microtubules align along the direction of the largest stress, with a protein severing those microtubules that are not aligned in this direction. As the stress inside a cell is determined in part by the cell's shape, this sets up a feedback loop: the stress resulting from the cell shape aligns the microtubules that reinforce the cell wall, thus maintaining the shape of the cell.

An external stress applied to the epithelium can override this internal stress. Because all of the plant cells are under turgor pressure from the inside, pressure from the outside, like squeezing a balloon, changes the stress pattern, causing the realignment of the microtubules so as to resist the new stress. This shows that the microtubules respond to local stresses within a cell, and are continually responsive to stress changes.

(*Pouille et al., 2009*). In *Arabidopsis* sepals, stochastic events including cell division and entry into endoreduplication also play a critical role in the distribution of cells of different shapes (*Roeder et al., 2010*). Altogether this suggests that cell behavior results from both local and supracellular cues. The exact role of such heterogeneity remains poorly explored, and how cells can differentiate between local and global cues is completely unknown. In this study, we show that mechanical stress act as a common instructing signal for microtubule (MT) orientation at both subcellular and tissue scales.

Mechanical forces have been proposed to provide directional information in control of MT orientation in plant cells and changes in mechanical forces are known to affect microtubule alignment (*Green, 1980*; *Williamson, 1990*; *Schopfer, 2006*). MT arrays have been proposed to align along maximal mechanical stress directions in the shoot apical meristem, as prescribed by tissue shape, assuming tension in the epidermis (*Hamant et al., 2008*). Mechanical forces were recently found to modify MT organization in leaf epidermal cell layers (*Jacques et al., 2013*). In *Arabidopsis* and most angiosperms, the cotyledon and leaf epidermal cells, also called pavement cells, exhibit typical jigsaw puzzle shapes, with indented regions and lobe-like outgrowths. The intracellular effectors of these morphologies are being described in many reports. In particular, indenting regions are enriched in cortical MTs, which are thought to restrain growth expansion via the presumptive localized deposition of stiff cellulose microfibrils (CMF) (*Fu et al., 2005*; *Yang, 2008*). Although this model seems relatively parsimonious, these biophysical assumptions have not been tested. The MT severing enzyme katanin is required for local MT ordering in pavement cell indenting regions, downstream of the plant hormone auxin and Rho GTPases (*Lin et al., 2013*). How robust shapes could derive from such regulation is however a subject of debate.

The complex morphology of pavement cells is a system of choice to decipher the contribution of cell and tissue shape-derived mechanical stresses in MT behavior. In this study, we have combined computational models and experiments to determine the relation between physical forces, material elasticity, and the behavior of cortical MT. We first relate MT behavior to cell wall reinforcements. Second, we confirm (in a different tissue than investigated in the past and at a different scale) that MTs orient along the predicted maximal tensile stress direction—and in this case, that they can do so at a subcellular or a supracellular scale, depending on the stresses involved. Lastly, we take advantage of the large size of the pavement cells to show how the MT response to stress depends on MT severing-dependent self-organization events. Altogether, this provides a scenario, in which not only tissue shape, but also cell shape, depends on a mechanical feedback loop. Based on our results, we propose that cells sense mechanical stresses at the subcellular scale, and that they are hence able to integrate cell shape-derived stresses and tissue shape-derived stresses, with a single mechanism.

## Results

### Pavement cell shape correlates with microtubule organization and consistent mechanical reinforcements in cell walls

The presence of parallel bundles of MTs in pavement cells is spatially correlated with indenting neck regions of the cell (*Figure 1A*, *Figure 1—figure supplement 1*; *Fu et al., 2005*). However, this correlation is debated, as MT orientations can be very noisy and pavement cell growth has even been proposed to be rather isotropic (*Zhang et al., 2011*). To quantify MT behavior in pavement cells, we used a nematic tensor-based tool to measure MT anisotropy (*Uyttewaal et al., 2012*; *Boudaoud et al., 2014*). This showed that MT arrays in indenting regions were more anisotropic than the MT arrays in lobes (mean ± SE is 0.40 ± 0.02 for indenting region and 0.20 ± 0.02 for lobes, n = 18 cells; 3 seedlings; p<0.01, *t* test; *Figure 1A,B*, *Figure 1—figure supplement 1*). Time lapse imaging of pavement cells showed that the anisotropy of MTs was maintained in indenting regions after 3 hr (mean ± SE is 0.47 ± 0.01, n = 14 cells; 3 seedlings), consistent with previous studies (*Zhang et al., 2011*) (*Figure 1A,B*, *Figure 1—figure supplement 1*). Note that we focused our analysis on a stage where cells have already attained their jigsaw puzzle shape and are still growing; our conclusions do not necessarily apply to these cells at a younger, undifferentiated stage or at an older, fully grown stage. The correlation between pavement cell shape and the presence of stably aligned MTs in indenting regions suggests that pavement cell shape relies on the impact of MTs on the mechanical anisotropy of the cell wall in these regions. This however has never been demonstrated. To directly measure mechanical properties of the outer wall with sufficient resolution, we used atomic force microscopy (AFM) to probe different regions of pavement cells, using an established protocol (*Milani et al., 2011*). First, we probed guard cells, in which the presence of transverse MT and CMF is unequivocally established (*Marcus et al., 2001*). The stiffness map of the outer wall of guard cells revealed the presence of transverse wall reinforcements matching the CMF orientation (*Figure 1C*). Therefore, our AFM approach has sufficient spatial resolution to reveal mechanical heterogeneities in cell walls. The AFM-based stiffness map of pavement cells also revealed the presence of mechanical heterogeneities, with apparent elastic moduli spanning the range 2–8 MPa. In particular, we found fibrous patterns of higher elastic modulus congregating at sites of indenting regions (*Figure 1D*, *Figure 1—figure supplement 2*). These lines are independent of small-scale topography (*Figure 1E*). Note that with AFM, we are measuring the mechanical properties of the wall in the normal direction to the wall surface with an isotropic probe, and therefore, we are not assessing the very local mechanical anisotropy of the cell wall in-plane. Yet, our data reveal a spatial pattern of mechanical anisotropy at a larger scale, and thus can be used as a proxy for mechanical anisotropy in sub-regions of the cell's outer wall. Staining for cellulose in mesophyll cells of wheat that exhibit alternating patterns of lobe like outgrowth and indenting regions has shown a strong presence of cellulose in the constricted regions of the cells (*Jung and Wernicke, 1990*). Altogether, this suggests that CMFs act as a brace around the indenting regions providing mechanical reinforcement of those regions.

Recent reports suggest that MT and cellulose synthase orientation could have different behaviors on the inner and outer face of hypocotyl epidermal cells (*Chan et al., 2010*; *Crowell et al., 2011*) and stems (*Fujita et al., 2011*). To test if this is the case in pavement cells, we visualized MT organization on the inner face of the pavement cells in confocal Z sections. The inner MT arrays appear to be identical to those observed on the outer face, thereby suggesting the cell walls of the indenting regions are reinforced on both faces of the cell (*Figure 1—figure supplement 3A,B*). These data thus consolidate a model in which pavement cell shape is maintained by localized MT-dependent wall reinforcements in indenting regions (*Fu et al., 2005*).

### Microtubule organization correlates with cell geometry-derived mechanical stresses

Next, we investigated whether mechanical stress can act as an instructive signal in this MT organization. Physical models of the shoot apical meristem as a pressurized vessel have shown that the supracellular mechanical stress pattern can prescribe global MT organization and tissue morphology (*Hamant et al., 2008*). Scaling down, this model implicitly predicts that mechanical cues also contribute to single cell morphology. However, this question still remains to be explored.

To address this, we first computed the expected patterns of stress in pavement cells using a three-dimensional (3D) finite element model representing the outer face of the wall as a curved

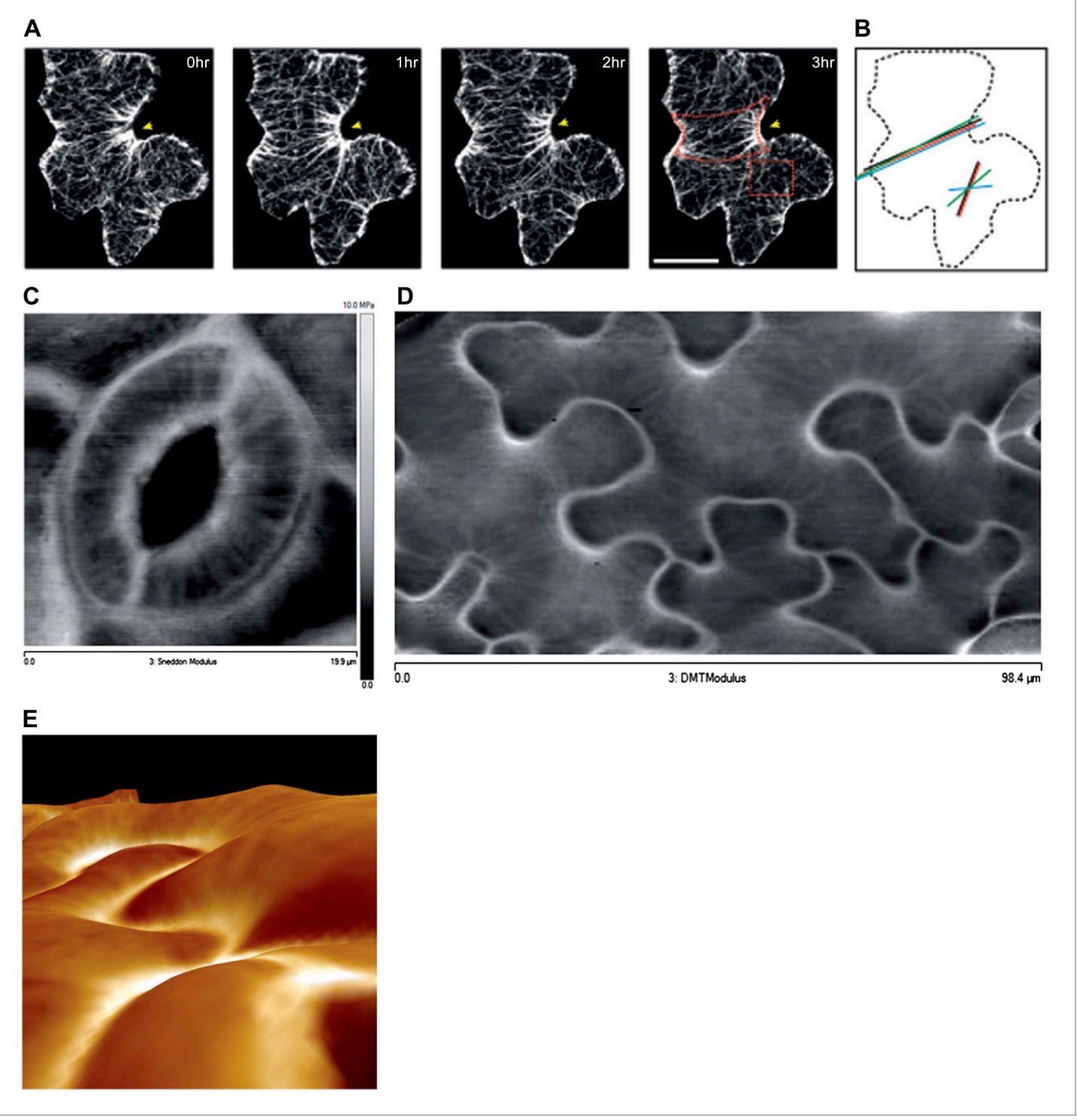

**Figure 1**. Mechanical heterogeneity of pavement cells correlates with microtubule patterns. (**A**) Microtubule bundling persists in indenting regions of pavement cells over time. Scale bars 20 μm. (**B**) Microtubule anisotropy over time, lines represent average orientation of microtubule arrays at different time points within the region of interest (dashed box in **A**). Longer lines indicate higher degree of anisotropy. (**C**) Stiffness map of the outer walls in two guard cells obtained by atomic force microscopy revealing transverse wall reinforcements. Gray scale bar represents scale of the observed elastic modulus in MPa. (**D**) Siffness map of the outer walls in pavement cells obtained by atomic force microscopy (same scale bar). Indenting regions exhibit striations with increased values of elastic modulus, reflecting regions with strong mechanical anisotropy. (**E**) 3D rendering of pavement cell topography as obtained by atomic force microscopy.
*Figure 1. Continued on next page*

*Figure 1. Continued*

The following figure supplements are available for figure 1:

**Figure supplement 1**. Microtubule organization and mechanical heterogeneity.

**Figure supplement 2**. Microtubule organization and mechanical heterogeneity.

**Figure supplement 3**. Microtubule organization and mechanical heterogeneity.

surface. Tensile stress experienced by the walls is caused by turgor pressure within the cells, therefore the main question was whether cell shape affects the anisotropy of stress in the wall. The 3D geometries of the cells were extracted from confocal microscope images of pavement cells processed with MorphographX and meshed with quadrilateral shell elements (*Kierzkowski et al., 2012*). The boundaries of the cells contain additional beam elements that account for increase in stiffness due to the presence of the anticlinal walls. For the material properties, we used a constitutive model of hyperelastic transversely isotropic material. To account for the mechanical anisotropy of the wall (*Baskin et al., 1999*), elasticity of the tissue was represented by the behavior of an isotropic matrix combined with the resistance of CMF oriented in a single-preferred direction per element. In the simulations, we have assumed that stress pattern arises due to turgor pressure in individual cells accompanied by tension in the epidermal layer (*Dumais and Steele, 2000*; *Kutschera and Niklas, 2007*; *Hamant et al., 2008*). The displacement of the anticlinal walls was restricted in z direction.

The outcome of the model with mechanically isotropic cell walls showed a strong anisotropic arrangement of stresses focused in indenting regions of the pavement cells (*Figure 2A,C*, *Figure 2— figure supplement 1A*). Despite the noisy behavior of MTs in pavement cells, visualization of MT organization with YFP:MBD (*Wightman and Turner, 2007*) showed a good correlation between the largest stress direction and MT arrangement, aggregating in indenting regions of the pavement cells (*Figure 2B,D*, *Figure 2—figure supplement 1B*). Furthermore, the clustering of MT arrays in the indenting regions also correlated with regions of predicted higher magnitude of stress (*Figure 1E*, *Figure 2—figure supplement 1C*). Stress directions were also computed in a model representing a 3D pavement cell shape along with the anticlinal and bottom walls. The model generated principal stress tensor direction pattern similar to the curved surface model (*Figure 2—figure supplement 2A–C*). Both cases showed a higher magnitude of tensile stresses in the indenting neck-like regions, matching the local MT pattern; compression forces at approximately 10% of the tension value were observed in some regions close to the cell boundary with no clear relation with MT behavior (*Figure 2—figure supplement 2D–F*). This suggests that tensile stress influences MT organization at a subcellular level in pavement cells, thereby indicating that the perception of stress must involve a mechanism that acts locally within each cell, rather than on a cell-wide or tissue-wide basis.

To check whether the mechanical anisotropy of the wall could alter stress direction in pavement cell models, we introduced a CMF direction with five times higher stiffness and subjected the cells to elastic expansion again. A feedback model between the stress perceived by a cell and orientation of the CMFs, which assumes an alignment of the CMF in the direction of maximal stress results in a very minor deviation of the stress pattern from the stress orientation obtained in the case of an isotropic material (*Figure 2—figure supplement 3*). The simulations thus suggest that for small elastic deformations the overall stress pattern is affected more strongly by the geometry of the cells than by the mechanical anisotropy of the existing cell wall material.

The model also predicts that the topography of the epidermis impact stress patterns and can have a dominant role on MT behavior as compared to cell shape. In particular, assuming that the epidermis is under tension, the largest stress direction should be circumferential around a local bump. Stomata being often in such an elevated position, we analyzed the MT arrays around guard cells and we found circumferential orientations, consistent with the predicted stress directions (*Figure 2F,G*). Conversely, no circumferential MT pattern could be observed near stomata that were not elevated (*Figures 2H,I*).

The results suggest that, analogously to what was observed in the shoot meristem, patterns of cellular and supracellular mechanical stress and MT orientation are correlated. Furthermore, we find that this correlation holds down to the sub-cellular level.

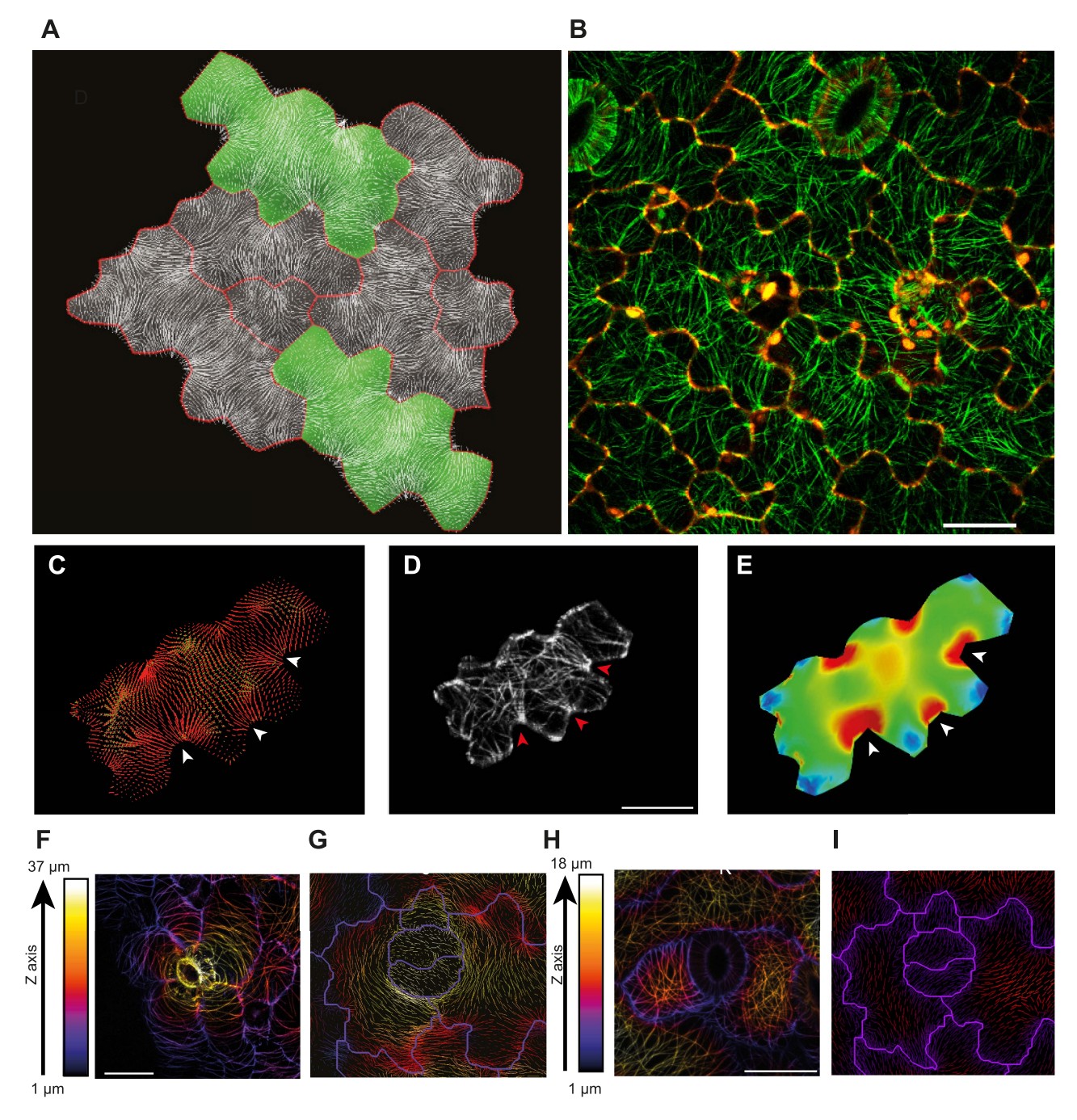

**Figure 2**. Microtubule patterns correlate with physical stress patterns. (**A**) Mesh showing stress directions, with the corresponding microtubule organization shown in panel **B**. Highlighted cells in green are represented in panels **C–E** and *Figure 2—figure supplement 1A–C*. (**C**) Largest stress direction (red) and second principal stress direction (green) in mechanical models of the pavement cell. White arrowheads indicate regions of convergence of directional tensile stresses in necks of pavement cells. (**D**) Microtubule (YFP-MBD) orientations correlate with the maximal stress direction predicted in the mechanical model. (**E**) Heat map showing the magnitude of stress distribution in the mechanical model. Arrowheads indicate regions of highest stress magnitude in neck regions. Scale bars 20 µm. (**F**) Circumferential distribution of microtubules surrounding elevated guard cells represented as a depth color-coded Z-stack. (**G**) Mechanical model of stress patterns around a stomata reproduce the observed arrangements of microtubules surrounding guard cells. (**H**) Microtubule organization around a non-elevated stoma and a mechanical model of stress patterns of the same (**I**). Scale bars 25 µm.

*Figure 2. Continued on next page*

*Figure 2. Continued*

The following figure supplements are available for figure :

**Figure supplement 1**. Microtubule organization and correlation with stress patterns.

**Figure supplement 2**. Simulation of single pressurized pavement cell shape A and D.

**Figure supplement 3**. Microtubule organization and correlation with stress patterns.

## Mechanical compression of cotyledon pavement cells affects microtubule anisotropy

Recent observations of MT organization in epidermal cells of leaves shows a supracellular response of MTs after changes in physical forces (*Jacques et al., 2013*). Application of compressive forces resulted in hyper-alignment of MTs. We tested whether mechanical stress can act as an instructive signal to organize MTs in cotyledon pavement cells by performing mechanical perturbations. Our finite element model predicts that when subjected to compression from above, there is an increase in overall stress in physical models (*Figure 3—figure supplement 1A,B*). Note that this response would however largely depend on the ability of the epidermis epidermal wall to maintain a constant volume while under compression. We directly applied compressive forces to pavement cells by using a coverslip that was pressed on the surface of the cotyledons and kept in place using adhesive silicone applied on the margins (*Figure 3—figure supplement 1C*). Depth color-coded Z stacks of MT organization and transects of the confocal Z-stack showed flattening of cells due to compression (*Figure 3—figure supplement 1C,D*). Imaging of MTs in the cells immediately after and 7 hr after compression showed rearrangement of MTs into more aligned arrays by 7 hr (*Figure 3—figure supplement 1C,D,F*; *Video 1*; mean ± SE is 0.37 ± 0.06 for 0 hr, n = 95 cells, 5 seedlings and 0.57 ± 0.06 for 7 hr, n = 95 cells; 5 seedlings; p<0.0001; Mann–Whitney U test; *Figure 3H*). This increased MT anisotropy persisted for longer periods when the tissue was maintained in the compressed condition (*Figure 3—figure supplement 2*). This response was also reversible: 24 hr following removal of compression, nematic tensor values for MT anisotropy were reduced to a value close to that obtained during the time point immediately after compression (mean ± SE is 0.24 ± 0.02; *Figure 3E,G,H*). It should be noted that the nematic tensor values between the 0 hr time point and the recovery state are not identical which could be the result of changes in imaging conditions ('Materials and methods'). However, despite this the reported nematic tensor values clearly show a trend in the increase and decrease of the MT anisotropy in the compression experiments. These results on cotyledon epidermal cells are thus consistent with the recently published report on contribution of mechanical stress in controlling MT behavior in leaf epidermal cells (*Jacques et al., 2013*).

## Large scale changes in tissue-wide stresses can override cell shape-derived stresses

A limit of the compression test is that water movements may be induced, and while it is extremely difficult to monitor, water flow may alter turgor pressure in the long term and thus the predicted stress pattern. So to further investigate the link between MT behavior and stress in pavement cells, we next induced large cuts to change stress patterns in cotyledons and observed the resulting MT pattern. The observation of tissue deformations after large cuts has long been used

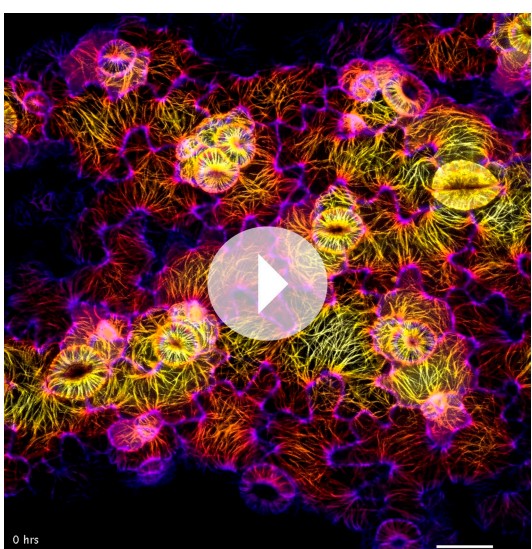

**Video 1**. Depth color-coded time series images showing changes in microtubule organization following compression. Scale bar 20 μm.

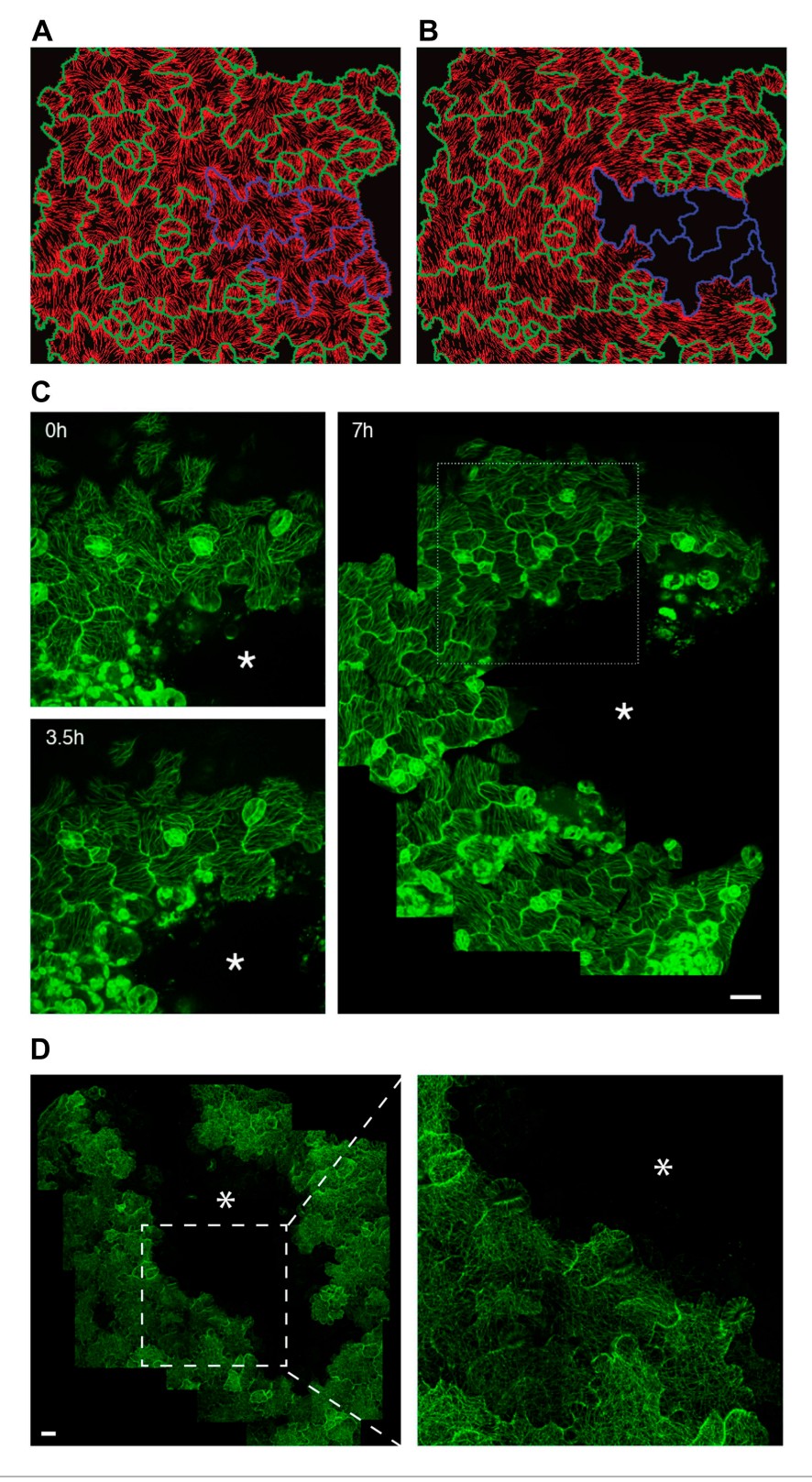

**Figure 3**. Extrinsic perturbation of mechanical forces induce directional changes in microtubule arrays. (**A** and **B**) Mechanical models showing changes in stress directions upon ablation. (**C**) Large scale ablation of cotyledons result in circumferential distribution of microtubule arrays around the site of physical perturbations, reproducing the

*Figure 3. Continued on next page*

*Figure 3. Continued*

results of the physical model. (**D**) Microtubule arrays in pavement cells of mutant *botero 1-7* shows random organization 7 hr after perturbation. Asterisk marks site of laceration. Scale bar 25 μm.

The following figure supplements are available for figure 3:

**Figure supplement 1**. Mechanical compression leads to increased microtubule anisotropy in pavement cells.

**Figure supplement 2**. Compression of pavement cells results in stabilization of microtubule array orientation.

**Figure supplement 3**. Cotyledon epidermis is under tension.

**Figure supplement 4**. Microtubule response to changes in physical forces in *katanin* mutant.

to deduce the stress pattern (***Dumais and Steele, 2000***) and has recently been adapted to calculate solid stresses in animal tumors (***Stylianopoulos et al., 2012***). Macroscopically, large cuts in cotyledons resulted in an immediate outward displacement of the cut edges, consistent with release of tension (***Figure 3—figure supplement 3A,B***; ***Video 2***). We also observed upward movement of mesophyll cells from the layer below (***Figure 3—figure supplement 3C***), consistent with the epidermis being under tension as observed in other plant tissues (***Kutschera and Niklas, 2007***). Simulation of such a large-scale nick in physical models was done with the assumption that laceration would lead to removal of turgor pressure from the cut cells followed by a reduction of wall elasticity, as previously published in more local (***Hamant et al., 2008***) and more global (***Dumais and Steele, 2000***; ***Kutschera and Niklas, 2007***) tissue contexts. Our physical model suggests that maximal stress directions become circumferential to the laceration after treatment and independent of cell geometry (***Figure 3A,B***; ***Video 3***). To test the model predictions, we performed similar lacerations on cotyledons of seedlings expressing fluorescent reporters of MTs. Time series imaging of MT rearrangements showed a progressive change in the organization of MT arrays around the site. After 3.5 hr, we observed hyper-aligned MT arrays in cells adjacent to the damaged region and at 7 hr the alignment advanced to cell layers farther away from the site of laceration, independently of cell shape (***Figure 3C***). Quantification of MT anisotropy showed a significant increase in MT anisotropy 7 hr after laceration (mean ± SE is 0.080 ± 0.007 for 0 hr, n = 61 cells and 0.37 ± 0.02 for 7 hr, n = 65 cells; 3 seedlings each; p<0.0001; Mann–Whitney U test), indicating that changes in tissue-wide stresses could override cellular level control of MT anisotropy.

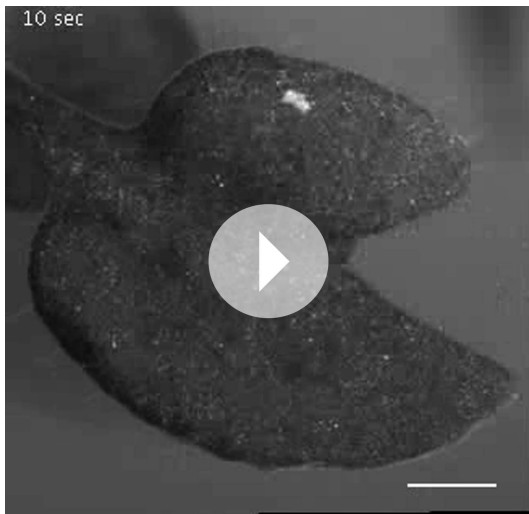

**Video 2**. Laceration of cotyledon shows outward displacement of cut edges. Scale bar 500 μm.

## Microtubule response to mechanical perturbation is dependent on the magnitude of change in stress intensity

To decipher if the response of MTs to mechanical perturbation depended on the intensity of stress imposed on the tissue, we performed simulation in which single cells were removed and the resulting patterns of stress computed. Similar to the large scale laceration experiments, a circumferential rearrangement of stresses was observed. However, the rearrangement was less pronounced than after cotyledon dissection, and observed in cells only adjacent to the site of perturbation (***Figure 4A,B***). To test this experimentally, we ablated single cells using a pulsed dye-coupled laser to see if small-scale perturbations could decouple the cellular level control of microtubule organization. 7 hr after ablation, we observed that MT alignment had responded to treatment by tending

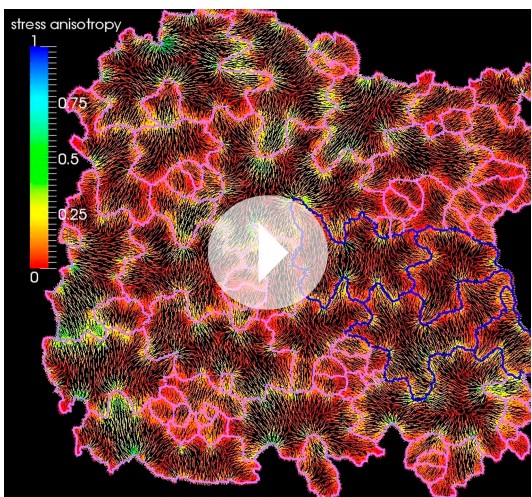

**Video 3**. Video of computational simulation showing circumferential distribution of stress and increase in MT anisotropy after ablation.

toward a circumferential alignment around the ablated cell site. The MTs around the site of ablation were not completely circumferential, consistent with a scenario in which stress anisotropy is weaker in single cell ablation cases than in cases of large-scale dissection, therefore leading to a comparatively reduced alignment of MTs around the site of ablation (*Figure 4C,D,E*). The alignment observed after single-cell ablation was significantly different from that resulting from large-scale perturbation, in which a strong circumferential response was obtained.

To further test if a directional force field could alter transverse MT organization observed in guard cells, we monitored MT arrays in guard cells adjacent to the region of large scale laceration. Single optical sections show that MT arrays still remain transverse in majority of the guard cells (*Figure 4—figure supplement 1*) indicating that these cells retain a certain degree of control of their MTs while being under the influence of a directional force field.

The observed rearrangement of MTs around the site of laceration supports our hypothesis but several other possibilities could exist. The changes observed could be due to a modified pattern of mechanical stress but also to the biochemical consequences of wounding of the sample. To investigate this we used the cellulose synthase inhibitor isoxaben, as a pharmacological means of increasing mechanical stress in pavement cells. Isoxaben was shown previously to induce hyperbundling and hyperalignment of cortical microtubules along the predicted directions of maximal stress in shoot meristem cells (*Heisler et al., 2010*), consistent with increased stress levels in the cell walls compared to non-treated plants. Treatment of 3-day-old seedlings with 40 μM isoxaben for 16 hr led to a sizeable increase in the anisotropy of microtubules (mean ± SE is 0.27 ± 0.04 for 0 hr, n = 53 cells and 0.54 ± 0.08 for 16 hr, n = 53 cells; 3 seedlings; p<0.0001, Mann–Whitney U test; *Figure 4—figure supplement 2A–C*). These observations add further evidence that mechanical forces are responsible for MT rearrangement. However, it should be noted that a much more complex scenario, involving both mechanics and wound responses, could govern this aspect of MT rearrangements.

## The microtubule response to subcellular and supracellular stresses depends on katanin-dependent severing activity

Homozygotes for the katanin loss-of-function allele *ktn1-3* exhibit severe pavement cell shape defects, consistent with the role of MTs in neck formation (*Lin et al., 2013*). When quantifying MT anisotropy in the *bot 1-7* katanin allele, we found a 60% reduction when compared to the wild type (Mean ± SE is 0.09 ± 0.01 for *bot 1-7*, n = 83 cells; 4 seedling and 0.23 ± 0.05 for WT, n = 53 cells; 3 seedlings), thus confirming the relation between MT ordering and pavement cell shape. It has been shown in the shoot meristem that the response of MTs to tissue stress relies on MT self-organization. In particular, MT rearrangement after changes in mechanical stress is promoted by katanin-dependent MT severing (*Uyttewaal et al., 2012*). To test whether, as in the shoot meristem, katanin activity is required for the MT response to stress in pavement cells, we performed lacerations and compressions in the katanin mutant background *bot1-7*. In both assays, MT arrays in pavement cells remained relatively isotropic after the micromechanical perturbations (*Figure 3D*, *Figure 3—figure supplement 4A–C*; *Video 4*). Measurement of MT anisotropy showed no significant difference before and 7 hr after ablation in *bot 1-7* (mean ± SE is 0.09 ± 0.02 for 0 hr, n = 83 cells and 0.10 ± 0.02 for 7 hr, n = 74 cells; 3 seedlings; p>0.05, *t* test; *Figure 3D*). Only subtle alignments could be detected in the immediate vicinity of the ablated zone in large-scale lacerations, but no alignment over several cell files was observed (*Figure 3D*). These results are consistent with observations made in the shoot apical meristem (*Uyttewaal et al., 2012*) and suggest that the MT response to stress is not only conserved in different tissues and at different scales, but also relies on a similar mechanism.

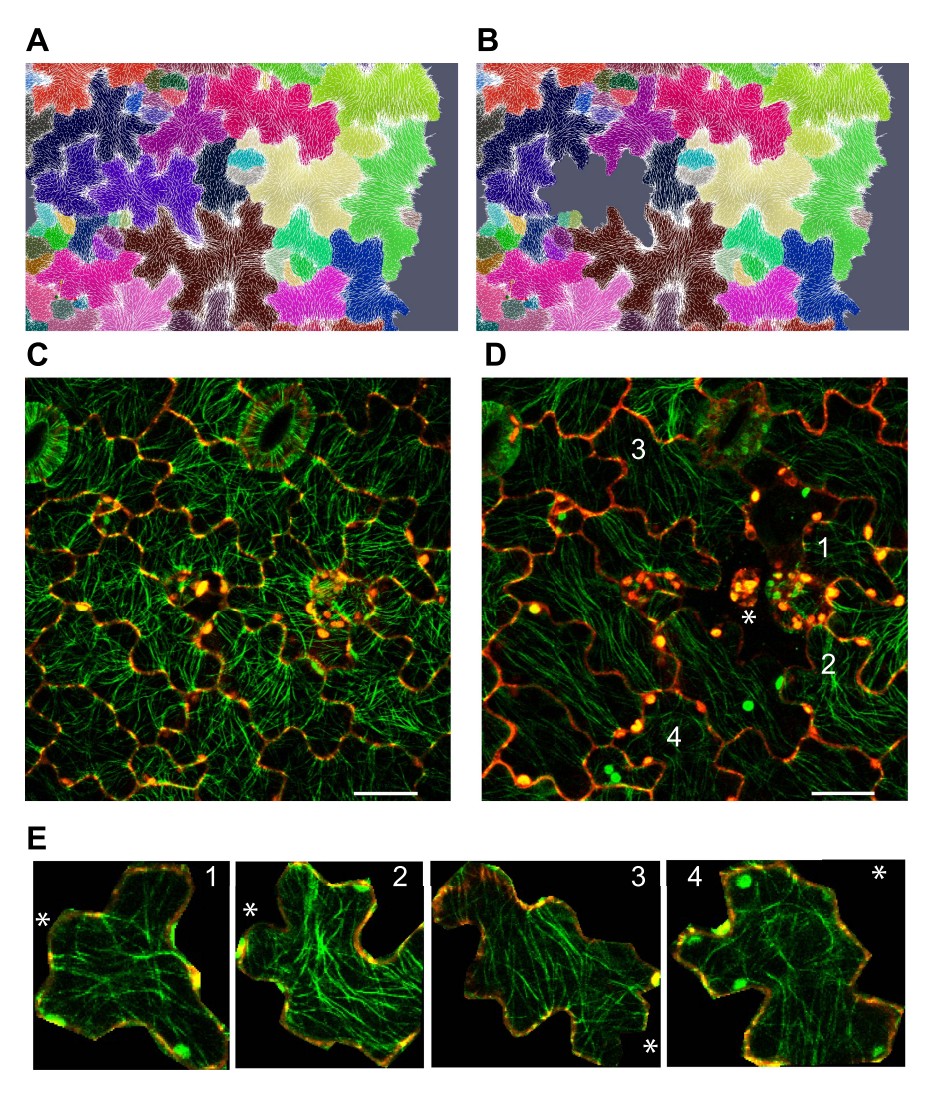

**Figure 4**. Stress intensity regulates microtubule alignment. (**A** and **B**) Simulation showing less pronounced circumferential rearrangements of stresses after ablation of single cell. Images of microtubule reporter line before (**C**) and 7 hr after (**D**) ablation of single cell, showing aligned microtubule arrays not completely circumferential after ablation of single cell. Scale bars 50 µm. (**E**) Magnified images of cells in figure **D** (Images not to scale). Asterisk shows the location of the ablated cell.

The following figure supplements are available for figure 4:

**Figure supplement 1**. Microtubule array organization in guard cells remains unaffected by changes in directional force field.

**Figure supplement 2**. Microtubule response to isoxaben treatment.

As quantification of severing events in meristems is impractical due to difficulties in accessing the cells for high-resolution imaging, we took advantage of the large size of pavement cells to further test this conclusion. To do so, we directly scored single MT-severing events in YFP-MBD cells adjacent to the lacerated region in pavement cells (*Figure 4A*). Using time-series imaging of MT arrays, we could observe severing of MTs occurring at sites of crossover points of MTs (*Figure 5A*; *Wightman and Turner, 2007*; *Lindeboom et al., 2013*; *Wightman et al., 2013*; *Zhang et al., 2013*). The severing rate in mock-treated control seedlings of $0.08 \pm 0.02 \times 10^{-3}$ events $\mu m^{-2}$ $min^{-1}$ (mean ± SD, N = 16 cells, 4 seedlings, total area of $2.99 \times 10^{4}$ $\mu m^{2}$) is in agreement with what has been previously published (*Wightman and Turner, 2007*; *Wightman*

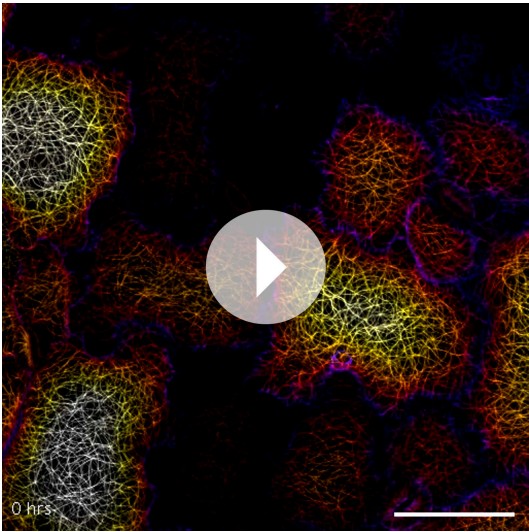

**Video 4**. Depth color-coded time series images showing microtubule arrays in *botero 1-7* does not induce hyper-alignment of microtubule arrays after compression. Scale bar 50 µm.

et al., 2013; *Zhang et al., 2013*). However, in seedlings subjected to changes in mechanical forces, cell files close to the site of ablation showed a significant increase in severing rate to $0.93 \pm 0.45 \times 10^{-3}$ events $\mu m^{-2}$ $min^{-1}$ (mean $\pm$ SD, N = 16 cells, 4 seedlings, total area of $1.5 \times 10^4 \mu m^2$, p<0.01, *t* test) 3 hr after ablation (*Figure 5B*; *Video 5*). Measurement of number of crossover points (i.e., two polymerizing MT crossing each other without experiencing severing or catastrophe) immediately after and 3 hr after mock treatment showed a 9% increase (mean $\pm$ SE is $11.0 \pm 1.4$ per 10 $\mu m^2$ for 0 hr, n = 6 cells and $12 \pm 1.5$ per 10 $\mu m^2$ for 3 hr) whereas the cells experiencing changes in mechanical stress showed a 65% decrease in the number of cross over points (mean $\pm$ SE is $15 \pm 3$ per 10 $\mu m^2$ for 0 hr, n = 6 cells and $5 \pm 2$ per 10 $\mu m^2$ for 3 hr). These measurements demonstrate an up-regulation in the severing of MTs at crossover sites after changes in mechanical stress. This mechanism could enrich the population of free MTs, while removing the MTs that do not align parallel to maximal tensile stress, thereby resulting in the generation of anisotropic MT arrays.

## Discussion

The role of mechanical stress in animals has mainly been investigated in single cells. Some pioneering studies in *Drosophila* have investigated the role of mechanical stress in tissues, and most notably in germ band extension (*Lecuit and Lenne, 2007*; *Sherrard et al., 2010*), dorsal closure (*Martin et al., 2009*; *Solon et al., 2009*), and gastrulation (*Pouille et al., 2009*) in relation to actomyosin reorganization. However the mechanical properties of these tissues have not been studied, notably because of the difficult physical access to these embedded tissues, and because the rapid growth dynamics of these tissues is not compatible with high resolution mechanical measurements, using for instance,

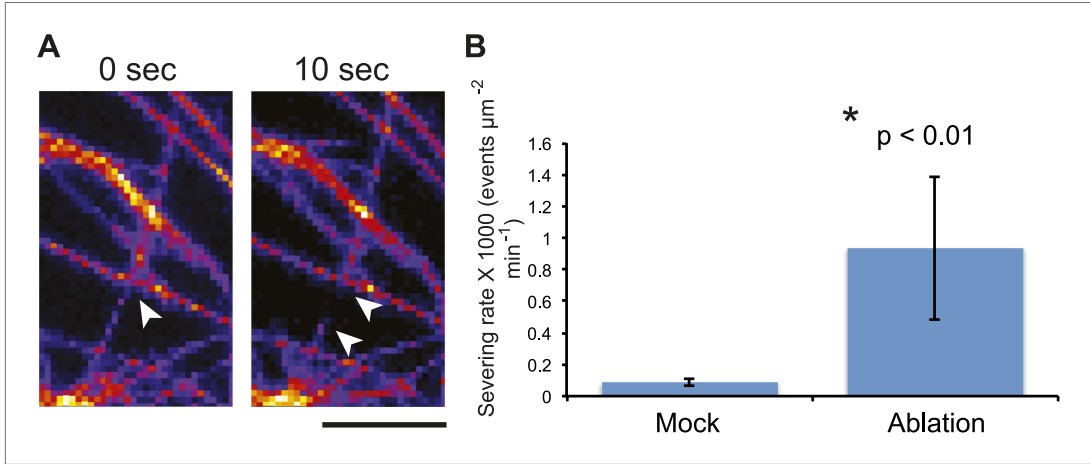

**Figure 5**. Mechanical perturbations increase bundling by promoting severing. 3D surface plot of YFP microtubule time series images representing a typical microtubule severing event (**A**), arrowheads indicate microtubule-severing at a crossover sites. Scale bar 5 µm. (**B**) Histogram representing microtubule severing rates of mock treated seedlings and in seedling of cells adjacent to site of ablation after 4 hours. Error bars represent standard deviation. Asterisk shows significance. (Student's *t* test). N = 16 cells, 4 seedlings, total area of $2.99 \times 10$ µm for mock and N = 16 cells, 4 seedlings, total area of $1.5 \times 10$ µm for ablation.

**Video 5**. Video showing severing of microtubule immediately after and 4 hr post ablation of cells. Red dots mark sites of microtubule severing. Scale bar 25 μm.

atomic force microscopy. Because plant growth is much slower and because tissues are easily accessible, such measurements are possible in plants. Here, we could correlate the behavior of the microtubular cytoskeleton with the predicted stress pattern and with quantified mechanical properties. This validates a number of previous studies assuming that pavement cell shape depends on mechanical heterogeneities (*Fu et al., 2005*), albeit relying only on MT behavior, and further supports plant tissues as facile systems to investigate the relation between the biophysics of growth and development.

Current models on pavement cell morphology suggest an influence of a ROP-based signaling mechanism on the cytoskeletal network (*Fu et al., 2005*; *Yang, 2008*). The findings suggest that growth is restricted in pavement cell indenting regions due to localized accumulation of anticlinal MT arrays by regulating CMF deposition, whereas local outgrowths are associated with the presence of an actin cytoskeleton that promotes growth. More recently it has been proposed that an auxin-dependent self-organizing mechanism controls the ROP-based signaling network (*Xu et al., 2010*). This model of differential growth is challenged by time lapse-imaging studies of pavement cell development, which shows an initial phase of multiple lobe initiations, followed by a phase of isotropic expansion during which the cell shape is maintained (*Zhang et al., 2011*). These studies do not take into account the mechanical aspects of the cell. In our study we show how subcellular mechanical stresses control MT organization, which in turn affects the mechanical anisotropy of the cell wall thereby contributing to cell shape irrespective of the previously proposed mechanisms. We believe that such a signaling module could add robustness to shape changes at the cellular scale.

Our studies further shed light on two theories that are at the center of many debates in biology. The organismal theory of multicellular organisms states that comprehension of tissue properties is essential to understanding the development of the organism, as opposed the view of cell theory, that proposes that the function of individual cells is what dictates development of the entire organism (*Kaplan, 1992*; *Baluska et al., 2004*). Previous findings on the shoot meristem suggest that MT organization depends on tissue shape-derived stresses (*Hamant et al., 2008*). Here, we found that cells can also interpret mechanical signals that are generated by their own shape. As this response is lost in large-scale mechanical perturbations, we propose that MT behavior depends on stress intensity, which is cell autonomous as long as tissue stresses do not override it. Such a balance between cell autonomous and non-cell autonomous stresses could control subcellular events in animal systems too.

Mechanical forces are known to cause changes to single MT dynamics in vivo by regulating the activity of MT-associated proteins (*Trushko et al., 2013*). In several organisms tensile forces acting on the kinetochore complex-attached MT ends are known to influence MT elongation rates during mitosis (*Nicklas, 1988*; *Skibbens et al., 1993*; *Gardner et al., 2005*; *Franck et al., 2007*). It remains to be tested if changes in mechanical forces could alter single MT dynamics in whole organism based studies. Previous studies on the shoot apical meristem show that katanin-mediated MT severing is required for MT response to changes in mechanical stresses (*Uyttewaal et al., 2012*). Our data not only show that a similar mechanism is present in pavement cells, but also show that severing itself is promoted by mechanical stress. One possible mechanism for stress-dependent regulation of MT alignment, where each MT is stressed due to their coupling with CMFs potentially by means of the cellulose synthesizing complexes which ride along the MT synthesizing the CMFs parallel to the MTs (*Paredez et al., 2006*; *Bringmann et al., 2012*). In such a scenario, a MT under tension is strained producing a conformational change to the MT lattice, which then could bias the binding of katanin to the less strained MTs at crossover sites, leading to preferential severing of the MTs that are not aligned to the principal stress direction. Reports have shown that conformation changes to the MT lattice act as hotspots promoting katanin binding and activity (*Davis et al., 2002*; *Díaz-Valencia et al., 2011*). As a result, this strain in MTs would lead to a MT alignment parallel to the anisotropic stress, not to overall cellular strain. MT behavior would thus depend on two parameters of mechanical stress (*Figure 6*): stress

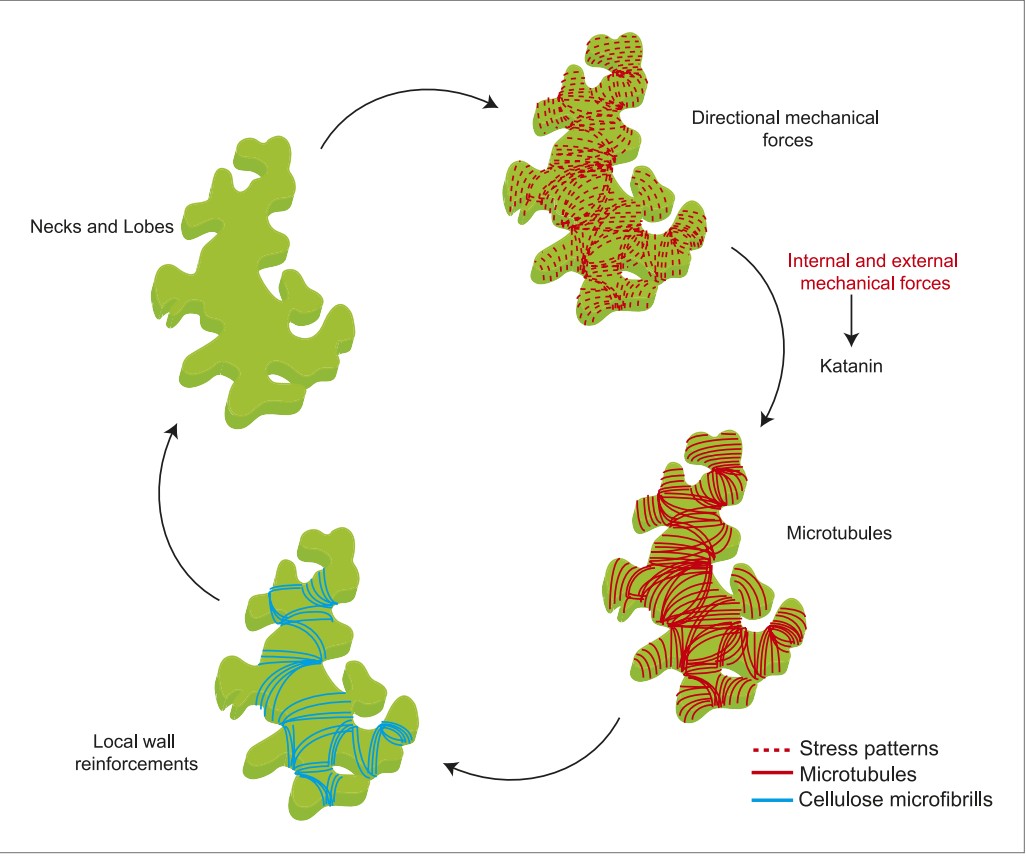

**Figure 6**. Mechanical forces regulate pavement cell shape by controlling microtubule organization and cellulose deposition.

direction from cell and tissue shape determine the dominant MT orientation, which dictates wall reinforcement, and thus in turn contributes to cell and tissue shape; and stress intensity modulates severing activity, which controls the response of MTs to stress. Based on these results, there is no need for several rules to explain MT behaviors at different scales or in different tissues in response to mechanical signals and perturbations, and this parsimony might be one of the defining features of mechanical signaling, when compared to molecular signaling. As shape rather relies on the actomyosin network in animals, the impact of mechanical forces on actin dynamics may in principle have a similar multiscale role in animals.

## Materials and methods

### Plant material and growth conditions

*Arabidopsis thaliana* lines expressing microtubule (MT) reporters YFP:MBD (Landsberg *erecta*) was previously used by *Wightman and Turner (2007)* and MBD:GFP (*Col-0*) was used by *Hamant et al. (2008)*. *botero 1-7* (*WS*) was isolated previously by *Bichet et al. (2001)*, YFP:MBD in a *botero 1-7* background and YFP:MBD (*Wassilewskija*) (*Wightman et al., 2013*) was a kind gift from Simon Turner (University of Manchester). All seeds were surface sterilized, stratified for 2 days, and grown vertically on plates containing half strength Murashige and Skoog (MS) media in light (16-hr photoperiod) at 21°C for 5 days for confocal microscopy.

### Specimen preparation and confocal imaging

5 days after germination plants were transferred to sterile plastic boxes containing MS media. The plants were fixed by adding lukewarm 1% agarose to the hypocotyls and roots submerging them, thereby exposing the cotyledons. Plants were imaged using a Zeiss LSM-780 or Zeiss LSM-700 as described in *Heisler et al. (2010)*.

## Mechanical perturbations

Laser-induced ablation was performed using an Andor Micropoint ablation laser fitted to a Zeiss LSM-780 scanning confocal microscope as described (*Hamant et al., 2008*) using a 63X or 40X water dipping objective. Compression was achieved by placing cotyledons tightly mounted between a cover glass and slide with silicon grease. Imaging of the compressed tissue was performed using an oil immersion lens. To image MT recovery after compression, seedlings were transferred to MS media containing Petri dishes after carefully removing the coverslip and treated as described above before imaging with a water-dipping lens. Laceration of the cotyledon was performed using a sharp forceps or a scalpel. Isoxaben treatment was preformed as described in *Heisler et al. (2010)*.

## Image processing

All images were processed and analyzed using the FIJI or ImageJ software. Background subtraction was performed using the 'Subtract Background' tool (rolling ball radius 30–40 pixels), and the 'StackReg' plugin was used to correct focal drift of the sample. Depth color-coding was performed using the 'Temporal-color code' tool. Cell boundaries in *Figure 2B* are pseudo projection of the lower most stack in a different color. Walking average function was performed on time series data sets used for analysis of severing events.

## Quantification of MT alignments for anisotropy

Quantification of MT alignments for anisotropy was done using the ImageJ macro described in *Uyttewaal et al. (2012)* and *Boudaoud et al. (2014)* for *Figure 1* and for the laceration experiments in which a score of 0 indicates completely isotropic pattern and 1, a case of completely aligned pattern. For all other experiments, the macro was modified in order to include in the output the visual and numerical measure of anisotropy of the nematic and texture tensors. The modified macro displays directions of both eigenvectors of the tensor of which relative length is an indication of the anisotropy of the signal. The score is calculated as a difference between the principal eigenvalues divided by the sum of diagonal elements of the tensor. The score gives 0 in case of completely isotropic pattern and 2 in case of completely aligned pattern. Analysis was carried out in each cell by drawing an outline of the region of interest, using the 'Polygon selection' tool in ImageJ. The outlines covered the entire area of the cell, without the anticlinal wall signal. These regions were then recorded using the 'ROI manager' tool in ImageJ and saved for automatic selection of the same cells in other time points.

## Severing quantification

Severing quantification was preformed manually using ImageJ, where severing events are scored using the point picker tool in ImageJ based on the criteria described in *Wightman et al. (2013)*. Quantification was carried out in cell files extending up to four layers from the site of ablation. A total of 15 events were observed in the mock treated samples in a total of 27.2 min and 66 events 3 hr after ablation in a total of 20.9 min.

## Statistical analysis

Statistics is preformed in Excel or OriginPro. For data sets that do not have a normal distribution, we have preformed Mann–Whitney U test, a non-parametric test of the null hypothesis; for normally distributed data sets, we have used the *t* test to estimate statistical significance.

## Atomic force microscopy

AFM indentation experiments were carried out with a Catalyst Bioscope (Bruker Nano Surface, Santa Barbara, CA), that was mounted on a optical macroscope (MacroFluo, Leica) using an objective (5x plano objective, Leica). To record surface topology and to create an elastic modulus map, PeakForce QNM AFM mode is used. A Nanoscope V controller and Nanoscope software versions 8.15 were utilized. All quantitative measurements were performed using standard pyramidal tips (ScanAsyst Air, Bruker, Inc.). The tip radius is given by the manufacturer to be between 2 nm and 10 nm. The spring constant of cantilevers was measured using the thermal tuning method (*Hutter and Bechhoefer, 1993*; *Levy and Maaloum, 2002*) and was ranged from 0.3–0.5 N/m. The deflection sensitivity of cantilevers was calibrated against a clean silicon wafer. All measurements were made under water at room temperature and the standard cantilever holder for operation in liquid was used. Cotyledons were detached from the stem with a fine blade and the abaxial (lower) side of the cotyledon was used for the measurement. The upper side of the sample was adhered to a Petri dish using a tissue section

adhesive (Biobond, Ted Pella, Inc.). Then the Petri dish was positioned on an XY motorized stage and held by a magnetic clamp. Then, the AFM head was mounted on the stage and an approximated positioning with respect to the cantilever was done using the optical macroscope.

The foundation of material property mapping with PeakForce QNM is the ability of the system to acquire and analyze the individual force curves from each tap that occurs during the imaging process. In this mode, the probe is oscillated at a low frequency (0.5 kHz), capturing a force curve each time the AFM tip taps on the sample's surface. The maximum force during imaging was 1 µN. For each sample, the topology and elastic modulus images were collected from different places of the sample over sizes of 70 × 70 to 150 × 150 µm$^2$ and at a digital resolution of 128 pixels × 128 pixels. The 0.3 Hz scanning rate was used. In this QNM mode technique, the elastic modulus is derived from the force–indentation curves by using 2 different models: (i) the Hertz–Sneddon model (*Sneddon, 1965*) or (ii) the Derjaguin-Muller-Toporov (DMT) model (*Derjaguin et al., 1975*). Both assumed a rigid cone indenting a flat surface. Unlike the Hertz–Sneddon model (*Equation 1*) that is applied on the approach curve, the DMT model is applied on the retract curve and accounts for adhesion (*Equation 2*).

$$F = \frac{2}{\pi} \frac{E}{(1-v^2)} \tan(\alpha)\delta^2 \tag{1}$$

*F* is the force from force curve, *E* is the Young's modulus, *v* is the Poisson's ratio, *α* is the half-angle of the indenter and *δ* is the indentation.

$$F - F_{adh} = \frac{4}{3}E^\star \sqrt{R(d - d_0)^3} \tag{2}$$

$F - F_{adh}$ is the force on the cantilever relative to the adhesion force, *R* is the tip end radius, and $d - d_0$ is the deformation of the sample. For this last model, the result of the fit is the reduced modulus $E^\star$. Moreover, if the Poisson's ratio is known, the software can use that information to calculate the Young's Modulus of the sample by the equation (*Equation 3*):

$$E^\star = \frac{E}{(1-v^2)} \tag{3}$$

Here, we assumed our sample is perfectly incompressible so that the Poisson's ratio used is 0.5. However, since neither the Poisson's ratio nor the tip shape is accurately known, we report in this work only an 'apparent modulus' (Ea).

## Computational models

For simulations of tissue mechanics, we have used a nonlinear Finite Element Method which provides an approximate solution of continuum elasticity problems on domains with complex geometry (*Zienkiewicz et al., 2005*). Simulations involving pavement cell geometries were performed with in-house written software, which is specialized and optimized for cellular geometries. The software is based on a procedure of minimization of nonlinear strain energy for a given constitutive model of the material (*Bonet and Wood, 1997*). We have used a hyperelastic transversely isotropic model of the material, which is particularly suited for description of fibrillar tissues (*Weiss et al., 1996*). For the matrix part of the material, we employed the Saint Venant–Kirchhoff model. The epidermal cell wall surface was modeled with shell elements, designed to handle thin curved surfaces. Additional beam elements accounting for the presence of the anticlinal cell walls were placed along the projections of the individual cell boundaries on the epidermal surface. These beam elements were assumed to have Young's modulus of the matrix part of the material and thickness of 1/5 of the epidermal wall. For the model on feedback between wall anisotropy and mechanical stresses we have set the material anisotropy axis at each step of the Newton–Raphson iteration aligned with maximal principal stress measured in the previous step. The model parameter values were chosen from different experimental estimates of plant cell wall elasticity based on measurements of in vivo samples (*Suslov et al., 2009*; *Hayot et al., 2012*; *Nezhad et al., 2013*) and synthetic biocomposites (*Chanliaud et al., 2002*). We set Young's modulus equal to 40 MPa for the matrix part of the material. The Young's modulus of the CMF was assumed to be five times larger than the matrix. Turgor pressure was set to 0.2 MPa and thickness of the epidermal wall was assumed to be 1 µm. We have also tested different values of these parameters varying them in the range 0.5–2 times the presented values and found the results of the simulations to be qualitatively consistent within these limits. To account for tension in the epidermal layer the outer boundary of the templates was

expanded by 1% in the x-y plane. The inner boundaries of individual cells were free in the x-y plane and restricted in the z direction. Pressure was applied on the surface of each cell. The choice of low turgor pressure was mainly dictated by convergence requirements of our model. Large turgor pressures caused instability of the model with respect to x-y movement of the cell boundaries, as a result of our approximate treatment of anticlinal walls. We have found, however, that the stress pattern does not show major qualitative changes for larger pressures. Thus, we have used the turgor within low range of experimentally reported values assuring good convergence with reasonable computational cost.

Most of the simulations involving the cell shapes extracted from experimental data were started from a flat geometry. The boundaries of the cells were obtained from the experimental data using MorphographX software (*Kierzkowski et al., 2012*) and the meshes constrained to those boundaries were constructed with the use of the CGAL algorithms (*CGAL, 2014*). For the stomata simulations (*Figure 2*), we have extracted information about the three-dimensional geometry of the cell surfaces using the three-dimensional confocal microscopy data and MorphographX software. In this case, the initial mesh was projected on the estimated real cell surface.

Simulations involving simpler geometries were performed with Abaqus (Simulia, Providence, RI) finite element modeling software (*Figure 2—figure supplement 2*, *Figure 3—figure supplement 1*), using the same material model as in other simulations and shell finite elements. These compression simulations were performed with a constant volume assumption and node to surface frictionless contact. Similar results were obtainable with a constant pressure assumption, but in that case the outcome of the simulation was dependent on material properties and the degree of deformation from pressing. To investigate the importance of the anticlinal walls on the stiffness of the epidermal wall model, we performed simulations with a simplified pavement cell shape including anticlinal walls in Abaqus (Dassault Systemes; *Figure 2—figure supplement 2*). The results were quantitatively consistent with our approximate model of the anticlinal walls. A detailed description of models is provided in supplemental information (*Supplementary file 1*).

## Acknowledgements

We would like to thank An Yan, Division of Biology, California Institute of Technology for his comments on the manuscript and Nathan Hervieux, ENS Lyon, for help with quantification of nematic tensor values in large scale laceration experiments. This work is supported by the Gatsby Charitable Foundation to HJ and EMM, the Department of Energy Office of Basic Energy Sciences, Division of Chemical Sciences, Geosciences and Biosciences, of the US Department of Energy [DE-FG02-88ER13873] to EMM, the Howard Hughes Medical Institute and the Gordon and Betty Moore Foundation (through Grant GBMF3406) to EMM, Agence Nationale de la Recherche ANR-10-BLAN-1516 'Mechastem' to OH, the European Research Council starting grant (PhyMorph, #307387) to AB and by the Swedish Research Council and the Crafoord Foundation to HJ.

## Additional information

### Funding

| Funder | Grant reference number | Author |
| --- | --- | --- |
| Howard Hughes Medical Institute and the Gordon and Betty Moore Foundation | GBMF3406 | Elliot M Meyerowitz |
| Gatsby Charitable Foundation | | Henrik Jönsson, Elliot M Meyerowitz |
| US Department of Energy | DE-FG02-88ER13873 | Elliot M Meyerowitz |
| Agence Nationale de la Recherche | ANR-10-BLAN-1516 | Olivier Hamant |
| European Research Council | 307387 | Arezki Boudaoud |
| Swedish Research Council | | Henrik Jönsson |
| Crafoord Foundation | | Henrik Jönsson |

The funders had no role in study design, data collection and interpretation, or the decision to submit the work for publication.

**Author contributions**
AS, PK, OH, Conception and design, Acquisition of data, Analysis and interpretation of data, Drafting or revising the article; RW, PM, Acquisition of data, Analysis and interpretation of data; AB, Acquisition of data; AB, Conception and design, Drafting or revising the article; HJ, EMM, Conception and design, Analysis and interpretation of data, Drafting or revising the article

## Additional files

**Supplementary file**
• Supplementary file 1. Additional information about computational models.

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
