## [Decision Letter]

Thank you for sending your work entitled “Subcellular and supracellular mechanical stress prescribes cytoskeleton behavior in Arabidopsis cotyledon pavement cells” for consideration at *eLife*. Your article has been favorably evaluated by a Senior editor, Detlef Weigel, and 2 reviewers, one of whom is a member of our Board of Reviewing Editors.

The Reviewing editor and the other reviewer discussed their comments before we reached this decision, and the Reviewing editor has assembled the following comments to help you prepare a revised submission.

Mechanical influences on developmental and cellular properties are an important topic of broad impact in plants and animals. After having been out of fashion for a few decades, this topic (aided by new tools) is experiencing a renaissance. This manuscript complements recent work on tissue-level dynamics in the plant shoot meristem by examining the relationships between growth, stress patterns and microtubules (MTs) in crenulated pavement cells of the leaf epidermis. New data here provide more convincing experimental support that MTs respond to stress, and novel work with an AFM system shows that MT patterns and cellulose MF patterns match. Exploiting the large size of pavement cells and using mechanical manipulations (ablations), the authors show a correlation between the frequency of katanin-mediated MT severing events and reactions to mechanical stress. Their observation that tissue level stress can override cell level stress was also documented in a recent paper from [26]. Altogether these experiments and models have the potential to have a broad impact on our thinking about mechanical influences on development.

There are a number of improvements that should be made to the models, and the manuscript could benefit from some clearer writing and some further analysis of data already collected. Specific major issues to be addressed in a revised manuscript are enumerated below.

1) Re-evaluation of models used to represent cells.

Reviewers were concerned about the models used to represent cells in this manuscript, specifically what appears to be a balloon-type model representing only the surface cell wall, whose stresses reported may not be representative of real pavement cells. When looking at the finer details of stress patterns of individual cells, the patterns do not look like what would be expected from a proper 3D model of a cell – the necks should be under compression and the lobes under tension (see for example [15] paper, which the authors cite). In the presented model all stresses are tensional, even after ablation, which may not be realistic. Moreover, the text is misleading in making this appear to be a 3D model when it may be a 2D surface of cells, tethered to a substrate by springs at the anticlinal walls. If it is too difficult or inappropriate to apply the 3D model based on their previous shoot apex work (where cells have a very different shape), please explain. Similarly, the model used (and assumptions) should be much more clearly explained in the main text, as the difference between a 2D model with springs and a full 3D model is potentially very important. It is a bit hard to justify making a fine-grained analysis of stresses in a situation where the geometry of the model itself has not been shown to be representative of the system. For example, in 3D one would expect the neck of a pressurized, isolated pavement cell to be under compression.

Minimal additions to the models include:

Begin by representing the compressional stresses in the model in a different color. If the neck regions are indeed under compression (at least after ablation of neighbor), this would be a satisfactory test of the utility of the model. A hand-drawn model with a few pavement cells in Abaqus would be OK. The cells do not have to be perfect, just qualitatively similar in shape (with lobes and necks). Then apply similar loading conditions as in previous simulations and compare it with the simulation where one cell is removed. Please show both tangential stresses, with a color scheme that also indicates compression.

Correct (or explain) why in Figure panels 4A and B (the model ablation) the cell outlines exactly overlap. Shouldn't the borders move a bit in B next to the cells that are ablated? What exactly does ablation change in the model, if not the boundary next to the ablated cells?

2) Dissect the cell vs. tissue response to stress

A major conceptual advance would come from a deeper analysis of the continuum of cell and tissue responsiveness to stress. The authors propose that “MT behavior depends on stress intensity, which is cell autonomous as long as tissue stresses do not override it”. It would be extremely valuable to calculate or model under what conditions a tissue-level stress would override a cellular stress.

Along these lines, the authors might test whether pavement cell lobes that differ in their orientation relative to the overall growth direction of the leaf and/or relative to the cut sites, have different sensitivities or degrees of MT anisotropies. The answer to this will give a parameter to address the cell autonomy vs. tissue response. Another simple test would be to look at whether guard cell MTs respond to cuts. Given the evidence that GCs created their own local zones of stress and oriented MTs, my guess would be no, but there might be an interesting correlation with “elevated” vs. planar and open vs. closed stomata with respect to responsiveness. This could provide another example of differential “set points” in the cell vs. tissue continuum.

Also, if cutting experiments release tension globally, then wouldn't the MTs be expected to become less oriented?

3) Formulate a compelling argument for how plant cells sense stress and how this generates shape.

The Abstract states “Force patterns in plant tissues control cell shapes...”. In support, the authors point to their (very nice) previous work that suggests that MT orientation is controlled by stresses in the tissue, which can be deduced from organ shape (Introduction, paragraph 2). However, this is difficult to reconcile even in the shoot apex. If cells start out with a random orientation creating isotropic growth, why doesn't the plant just create a globular structure? How can MTs be responsible for both tissue shape and cell shape if their orientation is primarily controlled by stress? Another interpretation would be that MT orientation is specified by a mechanism other than stress. Genetically defined organizing centers could control tissue and cell polarity, controlling the orientation of MTs. The shape change induced by the MT orientation would then direct the cellulose orientation, and thus cell wall anisotropy, and create non-trivial shapes. The resulting shape would then have a stress pattern that correlates with the MT orientation, which the authors observe. What is the mechanism for sensing stress? It is relatively easy to sense strains, but how can you sense stress, without invoking strain at some point?

4) Reword section on cell-wall anisotropy to reflect indirect measure of this property.

The claim that pavement cell shape relies on oriented MTs affecting cell wall anisotropy has not actually been clearly demonstrated. The AFM results that show the technique can see the CMF bundles in walls. The results are very nice and convincing, and it is believable that you are indeed making a map of the CMF network with the AFM. However this are not measuring cell wall anisotropy in-plane with these experiments as the text seems to claim. However I see nothing wrong with using this data as a proxy for cell wall anisotropy. One should also note that this still does not show a causal relationship between MT and CMF orientation in this system, but it is convincing correlative data given what is already know about the interaction there.

5) Separate the effects of material properties from those that arise from cellular geometry.

The authors propose that lack of movement of the necks after ablation proves that they are stiffer in this region (second paragraph of the Results section entitled “Pavement cell wall shape correlates with microtubule organization…”). However, intuitively, effects of geometry would dominate stiffness here. I would be surprised if the ablation of a neighbor cell in a model with uniform material properties would show an equal displacement in necks and lobes, and this should be tested.

6) Revisit cell-pressing experiments with better images and proper citations of previous literature.

It is difficult to see the differences in the experiments were cells are pressed by slides (Figure 3). In the text the differences are reported to be 0.37 vs 0.57 by their anisotropy measure. They report the same amount of difference (0.2) for necks vs lobes where the orientation is obvious. Why is it so much harder to see here? Also it is claimed that the effect is reversible, and that the value goes to 0.24 a day after the slide is removed. But this is quite a bit less than the starting point of 0.37, unless 0.13 of the difference appears instantly. If this is the case, the “before” condition needs to be quantified and presented for comparison. [26] have done similar experiments pressing on leaf epidermal cells; these previous results need to be mentioned and evaluated more extensively in this manuscript.

7) A devil's advocate position would be that the stress sensing mechanism the authors observe is just a secondary role for MTs in case of damage, etc. and that their orientation is primarily determined by other factors (i.e., geometry, genes, gradients, etc.). The authors should point to data to rule out this hypothesis, or an explicit statement that this position cannot be ruled out should be made.

[Editors' note: further clarifications were requested prior to acceptance, as described below.]

Thank you for resubmitting your work entitled “Subcellular and supracellular mechanical stress prescribes cytoskeleton behavior in Arabidopsis cotyledon pavement cells” for further consideration at *eLife*.

Your revised article has been favorably evaluated by a Senior editor, Detlef Weigel, and two reviewers, one of whom is a member of the Board of Reviewing Editors. We appreciate the careful consideration of review comments and the new models included in the rebuttal letter. Some of the data provided only in the rebuttal letter we felt would be useful to the readers of the manuscript and so we ask you to make a few minor revisions before acceptance, as outlined below:

1) There is one point regarding the modeling that should be addressed more directly in the paper, and not just the rebuttal, and that is the choice of low turgor pressure (2 bar) in the models. This was used to explain why there wasn't more movement in the model after ablation, and why this level was necessitated by their model (if it is higher they get larger deformations and the models do not converge).

A more realistic model seems simply not to be possible with the technology available and our expert reviewer agrees that that a more realistic model would probably behave in a qualitatively similar way, and thus would not change their results. However, it is important for the reader to be aware why the low turgor choice was made, the difficulties in model convergence in such a system, and the issue with anticlinal walls. This information should be in the paper in either the main text or as supplemental material, not just in the rebuttal letter.

2) The experiments in which single cells were ablated (Figure 12 and in revised text) and in which guard cell MTs were monitored after ablation (Figure 13) were quite informative and the latter should also be included in the manuscript in either the main text or as supplemental material.

---

## [Author Response]

*1) Re-evaluation of models used to represent cells*.

*Reviewers were concerned about the models used to represent cells in this manuscript, specifically what appears to be a balloon-type model representing only the surface cell wall, whose stresses reported may not be representative of real pavement cells. When looking at the finer details of stress patterns of individual cells, the patterns do not look like what would be expected from a proper 3D model of a cell – the necks should be under compression and the lobes under tension (see for example*
[15]
*paper, which the authors cite). In the presented model all stresses are tensional, even after ablation, which may not be realistic. Moreover, the text is misleading in making this appear to be a 3D model when it may be a 2D surface of cells, tethered to a substrate by springs at the anticlinal walls. If it is too difficult or inappropriate to apply the 3D model based on their previous shoot apex work (where cells have a very different shape), please explain. Similarly, the model used (and assumptions) should be much more clearly explained in the main text, as the difference between a 2D model with springs and a full 3D model is potentially very important. It is a bit hard to justify making a fine-grained analysis of stresses in a situation where the geometry of the model itself has not been shown to be representative of the system. For example, in 3D one would expect the neck of a pressurized, isolated pavement cell to be under compression*.

In general, for structures of complex shapes under complex loading and boundary conditions, it is difficult to predict a pattern of stresses. In simpler cases (like a pressurized, rotationally symmetric Mexican hat shape as in [15]) one can infer stresses from curvature of the surface, and in our simulation of a hat shape we also observe compression in some regions of this structure (similar to [15]). While Dumais and Steele report circumferential compression in more simple dome-shaped structures, we tend not to see this in our simulations ([22], Bozorg et al. (2014)). We can get a decrease in circumferential stresses. However, we mainly ascribe this to boundary effects, since if we smoothly continue the surface into a cylindrical shape we have tensional circumferential stresses throughout the tissue. In more complex structures, this is dependent on the precise geometry, boundary and loading conditions, especially for doubly curved surfaces (Figure 7, Bozorg et al (2014)).Author response image 1.Simulation of pressurized doubly curved, saddle-like shape.The color map shows the value of the second principal stress. Blue-low, red-high, black-values below zero. The region in the valley close to the boundary is under compression, which transforms to tension at the top as the curvature changes. This demonstrates the complex stress pattern depending on the geometry, loading and boundary conditions.

Furthermore the shapes described in Dumais and Steele are not comparable to those of pavement cells. Their model of a sunflower capitulum is a continuum surface, whereas in our cellular model of pavement cells we encounter discontinuities at the boundaries of the cells.

We have clarified the description of the model assumptions in the text. The model concerns a curved epidermal wall surface in 3D, which is where we experimentally observe microtubules and from where we obtain the geometry. The sub-epidermal walls are not considered explicitly, but their effect is taken into account by the boundary conditions at the cell borders. The turgor is applied to individual cells.

For simulations we developed FEM software that allowed us to obtain the geometry of cells directly from experimental pictures. We used shell elements, which are specifically designed for thin curved structures. These elements use surface description of geometry, but take into account the thickness as well. Previously (22) we used volumetric (hexahedral) elements in our model of meristem. These elements are of general applicability, but to maintain acceptable dimension ratios large numbers of them must be used for a description of thin wall structures. For pavement cells this restriction led to extensive computational cost and we argue that shell elements are the much better choice here. At present, however, representation of T-junction shells in our software is not accurate and this is another reason why we did not model geometry of anticlinal walls explicitly.

To address the reviewers comment we performed simulations in Abaqus (Dassault Systemes) comparing behavior of models with and without anticlinal walls present. Despite some quantitative differences, the pattern of principal stress, which is the measure in which we are most interested in our model, is qualitatively similar in all cases. The first principal stress is tensional everywhere (Figure 8). In particular in the neck regions the dominant stress is directed towards boundary. The second principal (tensional) direction is parallel to the boundary. The second principal stress becomes slightly compressive (∼10 % of tension value) in some regions close to the cell boundary in the lobes (Figure 8). We did not find any clear correlation between compressive forces and microtubule orientation. The models are consistent with our postulated relation between the direction of maximal tension and MT orientation.Author response image 2.Simulation of single pressurized pavement cell shape.A and D. 3D cell with epidermal, bottom and anticlinal wall. B and E. The cell model with bottom wall removed and replaced with boundary conditions at the bottom of anticlinal walls. C and F surface model of epidermal wall only with anticlinal walls replaced by boundary condition. A to C. The color map shows the value of the second principal stresses. Blue to red mark positive values (tension). The black represents negative values (compression). D to F. The color map shows (blue to red) shows the value of the first principal stresses. No compression is observed.

In summary, we acknowledge the approximate character of our model of pavement cell epidermal walls, but we do not think that the types of potential discrepancies that might be seen with more detailed models affect our conclusions.

*Minimal additions to the models*
*include:*

*Begin by representing the compressional stresses in the model in a different color. If the neck regions are indeed under compression (at least after ablation of neighbor), this would be a satisfactory test of the utility of the model. A hand-drawn model with a few pavement cells in Abaqus would be OK. The cells do not have to be perfect, just qualitatively similar in shape (with lobes and necks). Then apply similar loading conditions as in previous simulations and compare it with the simulation where one cell is removed. Please show both tangential stresses, with a color scheme that also indicates compression*.

Our assumptions of the ablation simulation were tension in the epidermal layer and loss of turgor pressure together with deterioration of the epidermal wall in the ablated cells. We were mostly interested in the stress pattern in outer walls and did not explicitly represent anticlinal walls. Now we have also performed the simulation for a few cells including their anticlinal walls in Abaqus (Figure 9). In either case we do not observe compressive stresses in the epidermal wall. Small compressions (an order of magnitude smaller than tension) are found in anticlinal walls in a few places close to the ablated region. The first principal stress direction is tensional and circumferential around the ablated region in the model with only the epidermal wall also in the simulation including anticlinal walls (Figure 9, cf. Figure 3 and Figure 3—figure supplement 4).Author response image 3.Simulation of ablation with 3D anticlinal walls included. The color map shows the value of first (A, C) and second (B, D) principal stresses (blue-low, red-high) before (A, B) and after (C, D) ablation. Red and yellow arrows (red: first principal stress, yellow: second principal stress). We observe circumferential rearrangement of main principal (tensile) stress around the ablation in the epidermal wall with some compressive (black in color scale) second principal stress in anticlinal walls.

*Correct (or explain) why in Figure panels 4A and B (the model ablation) the cell outlines exactly overlap. Shouldn't the borders move a bit in B next to the cells that are ablated? What exactly does ablation change in the model, if not the boundary next*
*to the ablated cells?*

The cell outlines in our ablation simulations do change. The observed shift in the stress pattern is a result of this change. The movement of the cell outlines is not very large, though, and not visible on the large scale of tissue. It can be observed in the close-up picture (Figure 10).

Since we are mostly interested in the stresses in the epidermal wall and our model treats anticlinal walls in a simplified manner we do not account for bulging of anticlinal walls in the ablation simulation.

More importantly, the ablation simulation in our model turned out to be quite unstable for large forces and displacements thus we remained in the small displacement limit, i.e., in a window where results could be analyzed. Regardless of the magnitude of the displacements, the simulations show the general principle by which the tension in the epidermal wall leads to circumferential pattern of stresses around the ablation, even in the presence of complex stress pattern due to the turgor pressure of individual cells.Author response image 4.Close up of a merged image of cell boundary movement before (white) and after (red) ablation.Scale bar 1 micrometer.

*2) Dissect the cell vs. tissue response*
*to stress*

*A major conceptual advance would come from a deeper analysis of the continuum of cell and tissue responsiveness to stress. The authors propose that “MT behavior depends on stress intensity, which is cell autonomous as long as tissue stresses do not override it”. It would be extremely valuable to calculate or model under what conditions a tissue-level stress would override a cellular stress*.

We agree that quantitative study of conditions at which tissue-level stress overrides cellular stress would be very valuable. This would require however the precise experimental measurements of the parameters used as an input to the simulations.

These include material properties, turgor pressure, stretch of the epidermal wall, and geometry of initial shape. At this point we have only approximate and general knowledge of these parameters estimated in the setups similar to our experiments.

That is why our conclusions are based on qualitative observations from simulations, for example the change in stress patterns resulting from changes in tissue surface geometry (Figure 2). We did find that these are not extensively dependent on the choice of parameters, suggesting that the gap between cell and tissue responsiveness to stress is large and thus that a dose-dependent response can be detected experimentally.

To test this hypothesis, we first generated a computational simulation of a single cell ablation. In such case circumferential re-arrangement of stresses is less pronounced and more limited in scope than in the large-scale ablation simulation, but still visible (Figure 11).Author response image 5.Simulation showing less pronounced circumferential rearrangements of stresses (white lines indicate maximal tensile stress directions) after ablation of a single cell (grey cell). **(A) Before ablation (B) After ablation.**

We observe a weak circumferential orientation of microtubules around single-cell ablation sites experimentally (see below and Figure 12). Although these data are qualitative, they clearly show that the response to ablation is dose-dependent: when the magnitude of stress is predicted to increase, microtubules become less sensitive to cell shapes and mainly align according to the supracellular stress pattern.

*Along these lines, the authors might test whether pavement cell lobes that differ in their orientation relative to the overall growth direction of the leaf and/or relative to the cut sites, have different sensitivities or degrees of MT anisotropies. The answer to this will give a parameter to address the cell autonomy vs. tissue response. Another simple test would be to look at whether guard cell MTs respond to cuts. Given the evidence that GCs created their own local zones of stress and oriented MTs, my guess would be no, but there might be an interesting correlation with “elevated” vs. planar and open vs. closed stomata with respect to responsiveness. This could provide another example of differential “set points” in the cell vs. tissue continuum*.

We agree with the reviewer that this is an interesting point. To test this we ablated single cells using a dye-coupled laser to see if small-scale perturbations could affect the cellular level control of microtubule organization. Seven hours after ablation we observed that microtubule alignment became independent of cellular geometry. However, the microtubules around the site of ablation were not completely circumferential, consistent with a scenario in which stress anisotropy is weaker in single cell ablation cases, therefore leading to weaker alignment of microtubules around the site of ablation (Figure 12). This observation is significantly different from the large-scale perturbation in which a strong circumferential response was obtained. Quantification in nematic tensor values of microtubule anisotropy near the site of a single cell ablation shows a mean value of 0.20 ± 0.06 (SE) as opposed to the 0.37 ± 0.02 (SE) in cells near large-scale ablations. Altogether, this supports our conclusion that microtubule response is dependent on stress intensity.Author response image 6.Images of microtubule reporter line before (A) and 7h after (B) ablation of single cell, showing aligned microtubule arrays not completely circumferential after ablation of single cell. Scale bars 50 μm. E Magnified images of cells in figure C (Images not to scale). Asterisk shows the location of the ablated cell.

As suggested by the reviewer we also observed microtubule arrays in guard cells close to site of large-scale ablations. Single optical sections show that microtubule arrays still remain transverse in majority of the guard cells (Figure 13) indicating that these cells retain a certain degree of control of their microtubules while being under the influence of a directional force field.Author response image 7.Close up image of microtubule arrays in guard cell before and after large-scale ablation.Majority of the guard cells retain the transverse pattern of microtubule arrays after laceration.

*Also, if cutting experiments release tension globally, then wouldn't the MTs be expected to become*
*less oriented?*

Cutting experiments change the pattern of stress by releasing it in mainly one direction, with much lower affect in other directions (for the tensional stress in the epidermis, they release it in one direction, but not in the orthogonal one) – so they change an isotropic pattern to one that is anisotropic, thereby causing microtubule alignment.

*3) Formulate a compelling argument for how plant cells sense stress and how this generates shape*.

*The Abstract states “Force patterns in plant tissues control cell shapes...”. In support, the authors point to their (very nice) previous work that suggests that MT orientation is controlled by stresses in the tissue, which can be deduced from organ shape (Introduction, paragraph 2). However, this is difficult to reconcile even in the shoot apex. If cells start out with a random orientation creating isotropic growth, why doesn't the plant just create a globular structure? How can MTs be responsible for both tissue shape and cell shape if their orientation is primarily controlled by stress? Another interpretation would be that MT orientation is specified by a mechanism other than stress. Genetically defined organizing centers could control tissue and cell polarity, controlling the orientation of MTs. The shape change induced by the MT orientation would then direct the cellulose orientation, and thus cell wall anisotropy, and create non-trivial shapes. The resulting shape would then have a stress pattern that correlates with the MT orientation, which the authors observe. What is the mechanism for sensing stress? It is relatively easy to sense strains, but how can you sense stress, without invoking strain at*
*some point?*

To start, we say nothing in this paper about how the shape of the pavement cells is generated. This is an interesting topic, but the experiments we have done to date are with pavement cells after they already are in their characteristic jigsaw puzzle piece shape. The initial generation of this shape would require elements that we have not yet studied. In the meristem, we could propose that tissue-derived stresses contribute to tissue shape emergence, because we analyzed the formation of young primordia and we had quantitative data on growth, from which we could argue that an initial differential growth would be sufficient to generate anisotropic stress, and thus a microtubule pattern, which in turn would promote organ outgrowth and organ separation from the meristem ([22], cf Bozorg et al. 2014) where different models are compared to explicitly show that a stress feedback promotes anisotropic organ growth). In pavement cells, as we did not analyze growth at a subcellular level, it is more cautious to focus our conclusion on cell shape maintenance.

Second, it is clear that other signals than stress can control microtubule orientation (e.g., light, see Lindeboom 2013). Yet, our data are consistent with mechanical stress playing a primary role in microtubule orientation. Could it be strain instead? Stress sensing mechanism will certainly involve an initial and very local measurement of strain (as that is the only way stress can be measured). There is however no convincing evidence that cell strain would act as a cue for microtubule orientation: the alignment of the microtubules is parallel to the principal direction of maximal stress both globally (22) and locally (this paper) but the alignment of the microtubules is not always correlated with the direction of maximal cell strain. In particular, microtubules tend to be perpendicular to the maximal strain direction in cells (e.g., in the peripheral zone of the meristem), whereas they are parallel to maximal strain in the boundary regions between shoot apical meristem and floral meristems, a region which is also under highly anisotropic tensile stress parallel to the microtubules (Burian et al. 2013, Bozorg et al. 2014). Thus a response to stress provides a more parsimonious explanation than a response to cell strain.

What sort of mechanism could account for this? To begin with, one of the primary findings of this paper are that the stress sensing mechanism is subcellular, and not at the whole cell level, as the microtubules in pavement cells are aligned to stress where it is anisotropic (at the necks) but not where it is isotropic, and in different directions within one cell, directions that correspond to the predicted principal direction of maximal stress in each different cell region where the stress is anisotropic.

One possible mechanism, then, one that we are proposing in the current version of the manuscript is that each microtubule is stressed (perhaps through its attachment to the cell wall cellulose fibrils – this indirect attachment could for instance be mediated by cellulose synthase complexes that ride the microtubules and project cellulose fibrils into the wall parallel to the microtubules Paredez et al. 2006; Bringmann et al. 2012). If a microtubule under tension is strained so as to make the severing enzyme katanin bind more poorly than to an unstrained microtubule, then severing will prefer MTs that are not aligned to the stress direction, and will eliminate them. This fits with the finding in the paper and in our previous work (56) that severing is necessary to achieve alignment upon mechanical perturbation. There is also evidence from the literature that katanin activity can discriminate between microtubule domains with different properties ([12]; Diaz-Valencia et al. 2011). In this way a strain in microtubules results in alignment parallel to anisotropic stress, not to overall cellular strain. There are certainly other possibilities (e.g., a role of tension-induced microtubule polymerization (see [55])), and this mechanism is not proven in any way, but this is an example where an initial strain imposed by a specific mechanism and specific subcellular complex could lead to alignment to stress.

We have therefore made the following additions in the discussion of our revised manuscript:

“One possible mechanism for tension-dependent regulation of MT alignment, where each MT is stressed due to their coupling with CMFs potentially by means of the cellulose synthesizing complexes which ride along the MT synthesizing the CMFs parallel to the MTs (Paredez et al*.* 2006; Bringmann et al*.* 2012). In such a scenario a MT under tension is strained producing a conformational change to the MT lattice, which then could bias the binding of katanin to the less strained MTs at crossover sites leading to preferentially severing only the MTs not aligned to the principal stress direction. Reports have shown that conformation changes to the MT lattice act as hotspots promoting katanin binding and activity properties ([12]; Diaz-Valencia et al. 2011). As a result, this strain in MTs results in a MT alignment parallel to the anisotropic stress, not to overall cellular strain.”

*4) Reword section on cell-wall anisotropy to reflect indirect measure of this property*.

*The claim that pavement cell shape relies on oriented MTs affecting cell wall anisotropy has not actually been clearly demonstrated. The AFM results that show the technique can see the CMF bundles in walls. The results are very nice and convincing, and it is believable that you are indeed making a map of the CMF network with the AFM. However this are not measuring cell wall anisotropy in-plane with these experiments as the text seems to claim. However I see nothing wrong with using this data as a proxy for cell wall anisotropy. One should also note that this still does not show a causal relationship between MT and CMF orientation in this system, but it is convincing correlative data given what is already know about the interaction there*.

We agree with the reviewers that by indenting in the normal direction to the wall with an isotropic probe, in principle, we could not deduce mechanical anisotropy in the wall in the orthogonal/periclinal direction. Yet, as we observe lines on the resulting stiffness map, this reveals the anisotropic nature of the cell wall at a large scale. In other words, each indentation point is not providing any information on the mechanical anisotropy of the wall, but the spatial pattern does provide this information. We also agree that the data remain correlative. We have clarified these two points in the revision.

*5) Separate the effects of material properties from those that arise from cellular geometry*.

*The authors propose that lack of movement of the necks after ablation proves that they are stiffer in this region (second paragraph of the Results section entitled “Pavement cell wall shape correlates with microtubule organization…”). However, intuitively, effects of geometry would dominate stiffness here. I would be surprised if the ablation of a neighbor cell in a model with uniform material properties would show an equal displacement in necks and lobes, and this should be tested*.

The referees are right that geometry can have an effect on the stiffness of the structure and thus the displacements. We performed a simulation in which a single isotropic pavement cell was pressurized, with part of its anticlinal walls fixed. This is similar to ablation of the surrounding cells on one side of the cell (right in our case). The results show that the displacement in such a situation is indeed not completely uniform (Figure 14). If we consider the deformation of epidermal wall only it is largest in the neck regions (Figure 14). If we include deformation of anticlinal walls we observe swelling of the lobed regions (Figure 14). We are thus thankful to the reviewer and in light of these new result that show that geometry can have a major impact on the cell deformation after ablation, we decided to remove this part from our manuscript: while the observed experimental result remains true and consistent with neck stiffening, it can depend on both cell geometry and neck stiffening, and therefore is too weak to support neck stiffening exclusively.Author response image 8.Color map shows total displacement (including z coordinate) in a single pressurized 3D pavement cell model.The left edges of the cell were fixed, simulating attachment to unaltered tissue. The right side of the cell is next to ablated region. The tips of the lobes in ablation neighborhood show low values of displacements (blue). The displacement of the necks is larger . The black outline shows initial position of the edges. A) Deformation of the epidermal wall alone. B) Deformation including anticlinal walls.

*6) Revisit cell-pressing experiments with better images and proper citations of previous literature*.

*It is difficult to see the differences in the experiments were cells are pressed by slides (Figure 3). In the text the differences are reported to be 0.37 vs 0.57 by their anisotropy measure. They report the same amount of difference (0.2) for necks vs lobes where the orientation is obvious. Why is it so much harder to see here? Also it is claimed that the effect is reversible, and that the value goes to 0.24 a day after the slide is removed. But this is quite a bit less than the starting point of 0.37, unless 0.13 of the difference appears instantly. If this is the case, the “before” condition needs to be quantified and presented for comparison.*
[26]
*have done similar experiments pressing on leaf epidermal cells; these previous results need to be mentioned and evaluated more extensively in this manuscript*.

We do agree with the reviewers that ideally, we should observe exactly the same values before compression and after recovery. Nevertheless, and despite these differences, the reported nematic tensor values clearly show a trend in the increase and decrease of the microtubule anisotropy in the compression experiments. The reason for the lower value and less clear images in the recovery phase is due to the change in the magnification factor and the different objectives used. The compressed state is imaged with a cover glass and higher NA objectives, whereas the uncompressed recovery state is imaged with a water-dipping lens having lower NA. This has already been explained in the Materials and Methods section in the previous version. This experimental setup could not be avoided because of the need to demonstrate the removal of cover slip to allow the surface geometry to recover to its curved state.

During the preparation of the manuscript, similar results were reported by [26]. For this reason we have moved this entire figure section to the supplementary data and more emphasis on [26] is given in that section.

*7) A devil's advocate position would be that the stress sensing mechanism the authors observe is just a secondary role for MTs in case of damage, etc. and that their orientation is primarily determined by other factors (i.e., geometry, genes, gradients, etc.). The authors should point to data to rule out this hypothesis, or an explicit statement that this position cannot be ruled out should be made*.

We do agree that the observed changes could be a result of other factors as suggested by the reviewer. However our claims are based on already published work on the shoot apical meristem in which similar perturbation was performed (22; 24; 56). Evidence of microtubule rearrangements are also shown under conditions where wounding (ablation) or a change in cell geometry (compression) are not induced: isoxaben was used as a pharmachological agent to mechanically reduce the strength of the cell wall thereby increasing tension (24). Treatment of pavement cells with isoxaben resulted in a hyperaligned microtubule array, supporting our claim.

However as suggested by the reviewer we cannot rule out the possibility of other factors. We have modified the text and included this in our text.

*[Editors' note: further clarifications were requested prior to acceptance, as*
*described below.]*

*We appreciate the careful consideration of review comments and the new models included in the rebuttal letter. Some of the data provided only in the rebuttal letter we felt would be useful to the readers of the manuscript and so we ask you to make a few*
*minor revisions before acceptance, as outlined below:*

*1) There is one point regarding the modeling that should be addressed more directly in the paper, and not just the rebuttal, and that is the choice of low turgor pressure (2 bar) in the models. This was used to explain why there wasn't more movement in the model after ablation, and why this level was necessitated by their model (if it is higher they get larger deformations and the models do not converge)*.

*A more realistic model seems simply not to be possible with the technology available and our expert reviewer agrees that that a more realistic model would probably behave in a qualitatively similar way, and thus would not change their results. However, it is important for the reader to be aware why the low turgor choice was made, the difficulties in model convergence in such a system, and the issue with anticlinal walls. This information should be in the paper in either the main text or as supplemental material, not just in the rebuttal letter*.

We agree with the reviewers’ comments and we have included this section in the section describing the computation models in the Methods part of the main text (“Computational models” section).

*2) The experiments in which single cells were ablated (Figure 12 and in revised text) and in which guard cell MTs were monitored after ablation (Figure 13) were quite informative and the latter should also be included in the manuscript in either the main text or as supplemental material*.

We have now included the results of single cells ablation as part of the main figures (Figure 4) and include the observation of microtubules in guard cells in the main text along with a supplemental figure (Results section entitled “Microtubule response to mechanical perturbation is dependent on the magnitude of change in stress intensity”, Figure 4—figure supplement 1).